

# Friend leukemia integration 1 overexpression decreases endometrial receptivity and induces embryo implantation failure by promoting *PART1* transcription in the endometrial epithelial cells

Yile Zhang[1,2], Beining Yin[2], Sichen Li[2], Yueyue Cui[2] and Jianrong Liu[1]

[1] Reproductive Medical Center, The Fifth Clinical Medical College of Shanxi Medical University, Shanxi Provincial People's hospital, Taiyuan, China
[2] Reproductive Medical Center, the First Affiliated Hospital of Zhengzhou University, Zhengzhou, China

Corresponding author
Jianrong Liu, liujianrong3@sina.com

## ABSTRACT

**Background**. *In vitro* fertilization-embryo transfer (IVF-ET) is a crucial assisted reproductive technology for treating infertility. However, recurrent implantation failure (RIF), a significant challenge in IVF-ET success, remains unresolved. This study aimed to explore the role and mechanism of FLI1 in endometrial receptivity and RIF.

**Methods**. Differential endometrial cell proportions between patients with RIF and control subjects were assessed using single-cell RNA sequencing (scRNA-seq) analysis. The chromatin accessibility of FLI1 in the luteal endometrial tissue of patients with RIF and control subjects was examined using the single-cell assay for transposase-accessible chromatin sequencing (scATAC-seq). FLI1 mRNA and protein levels were gauged by quantitative real-time polymerase chain reaction (qRT-PCR) and western blotting. Cell viability and migration were examined via cell counting kit (CCK)-8 and scratch healing assays. Epithelial-mesenchymal transition markers were analyzed using western blotting. Mechanisms underlying FLI1's regulation of PART1 transcription and expression in endometrial epithelial cells were explored using chromatin immunoprecipitation and dual-luciferase reporter assays. Adeno-associated virus (AAV) carrying epithelial cell-specific FLI1/PART1 overexpression sequences was uterinely injected in mice to assess FLI1/PART1 effects.

**Results**. scRNA-seq revealed diminished endometrial epithelial cell proportions in RIF patients. Meanwhile, scATAC-seq indicated enhanced chromatin accessibility of FLI1 in these cells. FLI1 exhibited specific expression in RIF patients' endometrial epithelial cells. Specific FLI1 overexpression inhibited embryo implantation, while knockdown enhanced it. Pregnant mice injected with AAV encoding FLI1 overexpression had significantly lower implantation than AAV-negative controls. FLI1 binding to PART1 promoter heightened PART1 transcription and expression in endometrial epithelial cells. Rescue experiments illustrated FLI1's role in embryo implantation by boosting PART1 expression. PART1 was notably elevated in RIF patients' luteal endometrial tissue and non-receptive endometrial epithelial cells (HEC-1-A). Specific PART1 overexpression dampened embryo implantation, whereas knockdown promoted it. Pregnant mice injected with AAV encoding PART1 had lower implantation than

negative controls. PART1 knockdown mitigated FLI1's inhibitory impact on HEC-1-A cell viability and migration.

**Conclusions**. FLI1 overexpression in the endometrial epithelial cells of patients with RIF inhibited embryo implantation by binding to the *PART1* promoter region to promote *PART1* expression. These findings can aid in the development of novel therapeutic targets for RIF.

## INTRODUCTION

Recently, assisted reproductive technology (ART) has been increasingly used in the treatment of female and male infertility. *In vitro* fertilization-embryo transfer (IVF-ET) is a commonly used effective therapeutic strategy for infertility. However, recurrent implantation failure (RIF) is a key limiting factor affecting clinical pregnancy rates. The etiology of RIF is complex and highly heterogeneous, including endometrial receptivity, embryo aneuploidy, male factors, thrombophilia, immunological changes, microbiome, and anatomical aberrations (*Franasiak et al., 2021*; *Ma, Gao & Li, 2022*). Aberrant endometrial receptivity is one of the key factors leading to RIF. Approximately two-thirds of implant failures are caused by decreased endometrial receptivity (*Moustafa & Young, 2020*). In addition to the cyclical changes in endometrium morphology, a complex network of factors, including gene expression, protein modification, and microRNA (miRNA) regulation, regulates endometrial receptivity and implantation (*Hernández-Vargas, Muñoz & Domínguez, 2020*; *Liang, Wang & Wang, 2017*; *Yang et al., 2022*). However, the regulatory mechanism of endometrial receptivity has not been completely elucidated. The critical cell fate decisions involved in this essential biological process occur at the individual cell level. Although cellular systems exhibit coordination, single-cell analysis can reveal variations in genomes, transcriptomes, and epigenomes during cell division and differentiation even within the same cell line or tissue. Recent technological advancements have enabled the analysis of gene expression profiles at the level of individual cells using single-cell RNA sequencing (scRNA-seq). This provides a unique opportunity to precisely define the cell types and molecular pathways involved in maintaining tissue homeostasis and disease progression (*Jovic et al., 2022*). Recent studies have used scRNA-seq technology to examine cellular composition and intercellular communication events in the human endometrium during the menstrual cycle (*Garcia-Alonso et al., 2021*; *Wang et al., 2020*). Single-cell transcriptome profiling analysis revealed that RIF is associated with the downregulation of $CD49a^+CXCR4^+$ natural killer (NK) cells and a subset of $CD63^{high}$ $PGR^{high}$ endometrial epithelial cells with high levels of progesterone receptor. Autophagy and exosomes are reported to decrease the subset of NK cells (*Lai et al., 2022*). This study subjected endometrial tissue to scRNA-seq to identify candidate genes specifically expressed in endometrial epithelial cells of patients with RIF.

The mechanism underlying the regulatory effects of candidate genes on endometrial receptivity and RIF was elucidated using cell and mouse experiments.

## METHODS

### Patient enrollment and sample collection

This study recruited 20 patients with RIF who had no clinically clear etiology in the RIF group. Additionally, 20 subjects who achieved clinical pregnancy through 1–2 embryo transfers were recruited in the control group. Patients in the control group underwent *in vitro* fertilization solely due to male factors. Patients of both groups were assisted with artificial pregnancy through *in vitro* fertilization/intra-cytoplasmic sperm injection-embryo transfer. In the RIF group, the characteristics of the patients were as follows: failed clinical pregnancy, underwent 2–3 transplants, cumulative number of high-scoring blastocyst embryos transferred was ≥4, or cumulative number of high-scoring blastocysts transferred was 2–3. To reduce interference and ensure the credibility of the results, patients with other medical conditions were excluded from this study. Additionally, patients who received hormone therapy or other uterine procedures in the last three months were excluded. The endometrial tissue was harvested at days 6–8 post-ovulation in the natural cycle (embryo implantation window). This study was approved by the Ethics Committee at the First Affiliated Hospital of Zhengzhou University (2020-KY-256). Written informed consent was obtained from all study patients.

### Single-cell suspension preparation

The endometrial tissue of patients was washed with pre-cooled phosphate-buffered saline (PBS) and incubated with ethylenediaminetetraacetic acid (EDTA) to reduce the stability of the cell membrane. Next, the endometrial tissue was digested with protease/collagenase to prepare the single-cell suspension. The mononuclear cells were prepared using single-cell nucleus separation solution and subjected to scRNA-seq.

### Library construction and RNA sequencing

Library construction and RNA sequencing were performed, following the previously reported methods (*Corces et al., 2017*). Briefly, 50,000 cells were washed with PBS and lysed. Next, the lysate (50 μL) was centrifuged at 800 g and 4 °C for 10 min to obtain cell nuclei. The samples were immediately subjected to the transposition reaction. The transposition reaction mix (including the reaction buffer, Nextera Tn5 transposase (the amount of enzyme was optimized), and ribozyme-free water) was prepared and incubated with the prepared cell nucleus with mixing at 37 °C for 30 min. The DNA was purified using the Qiagen purification kit and eluted to obtain 10 μL of eluate. The transposable DNA was amplified using polymerase chain reaction (PCR) (first a pre-PCR was performed to evaluate the number of PCR cycles). The amplicons were purified and subjected to quality control using Bioanalyzer. The library was quantified using the Kapa library quantification kit (observe the presence of a typical nucleosome distribution pattern). High-throughput sequencing was performed using a computer.

## Generation of single-cell gene expression matrices

Each nucleus of the qualified sample was mixed with reagents and magnetic beads with barcode sequence successively on the 8-channel microfluidic chip and covered with oil drops to form gel beads in extrusion containing single nuclei and single gel beads. Each gel bead carries several sequences that are combined with Illumina connector sequence (P5), 16 bp 10x cell barcode (Barcode), and some Illumina read1 primers.

The constructed library was qualified using Qubit 2.0. The insert DNA of the library was examined using Agilent Bioanalyzer High Sensitivity DNA chip. After the expected size of the insert was obtained, the effective concentration (2 nM) of the library was accurately quantified using quantitative real-time PCR (qRT-PCR) analysis to ensure the quality of the library. Next, sequencing was performed using the Nextseq 500 platform.

## Quality control, cell type clustering, and major cell type identification

To ensure high-quality data, offline data quality control measures, such as the verification of sequencing quantity, barcode carrying rate, and sequencing quality Q30 value, were employed. Additionally, the barcode and sample index sequences were subjected to Q30 quality control to ensure high resolution at both the cell and sample levels.

The nucleosome, which comprises DNA and histone, is the basic structural unit of a chromosome. Each histone octamer comprises two histone molecules with approximately 200 bp of DNA sequence coiled around the octamer core structure to form a nucleosome. In this 200-bp sequence, the 146-bp DNA sequence coils directly around the histone octamer core and is resistant to nuclease digestion, while the remaining DNA sequence links the nucleosome to the next nucleosome. Nucleosome localization refers to the precise determination of nucleosome location on the genome, which is always accompanied by gene transformation from inhibition to transcription. Nucleosome localization, which has a major role in transcriptional regulation, DNA replication, and repair, is currently a research hotspot in epigenetic research. Based on nucleosome structural characteristics and the distribution of ATAC-seq inserted fragments, fragments within 147 bp of inserted fragments are designated as nucleosome-free regions, while fragments with a size of 147–294 bp are considered nucleosome distribution regions, providing accurate nucleosome localization.

The size distribution of ATAC-seq insertion fragments provides information on the packaging and location of nucleosomes, while the fragment length captures the periodicity of nucleosome positioning. Fragments with a size less than 147 bp indicate nucleosome-free regions, whereas those with a size of approximately 147–294 bp suggest a single nucleosome distribution region. The length and quantity of insertion fragments periodically vary due to the nucleosome structure. The fluctuation frequency is correlated with the length or number of nucleosomes.

To evaluate the number of cells captured in a barcode-labeled multi-cell mixed sample sequencing library, peak calling and insertion fragment data alignment to peak regions were performed. The number of cells successfully captured was determined by evaluating the insertion fragment count and length covering the peak region. Based on the barcode and corresponding insertion fragment count statistics, cells are classified into cellular and
non-cellular categories. Cells that exceed the threshold of captured insertion sequences are classified as single cells, while those with insufficient captured insertion sequences are classified as non-cells.

The simultaneous sequencing of libraries from multiple cells was performed by 10x Genomics to obtain the ATAC data for multiple cells. The Cell Ranger ATAC package uses the Python library MOODS (https://github.com/jhkorhonen/MOODS) to scan for peak matches with group positions in each cell and annotate peaks using the JASPAR database to group cells. Two-dimensional cell clustering results are obtained using the t-distributed stochastic neighbor embedding (t-SNE) dimensionality reduction algorithm.

## qRT-PCR analysis

Freshly collected endometrial tissue from the implantation stage was transferred to a culture dish, cut into pieces, and digested with collagenase at 37 °C for 30 min. The digestate was filtered using a filter to remove endometrial stromal cells and obtain primary endometrial epithelial cells. Total RNA was extracted from the samples. The isolated RNA (500 ng) was reverse-transcribed into complementary DNA using the HiScript II SuperMix (Vazyme, Beijing, China). qRT-PCR analysis was performed using the SYBR green master mix with the ABI 7500 System (Thermo Fisher Scientific, Waltham, MA, USA). The PCR conditions were as follows: 46 cycles of 94 °C for 10 min, 94 °C for 10 s, and 60 °C for 45 s. *GAPDH* was used as an internal reference.

## Western blotting

The protein samples were subjected to sodium dodecyl sulfate-polyacrylamide gel electrophoresis using a 10% gel. The resolved proteins were transferred to a polyvinylidene difluoride membrane. The membrane was blocked with 5% skimmed milk powder solution for 1 h and incubated with the following primary antibodies overnight: anti-FLI1 (554266; 1:500, BD Biosciences, rabbit anti-vimentin (ab92547, 1:1,000, Abcam), rabbit anti-E-cadherin (ab40772, 1:1,000, Abcam), and anti- $\beta$-actin (#4970L, 1:5,000, Cell Signaling Technology) antibodies. Next, the membrane was incubated with secondary antibodies for 2 h. Immunoreactive signals were developed using enhanced chemiluminescence.

## Source and analysis of long non-coding RNA (lncRNA) Data

The GSE26787 and GSE111974 datasets, which comprised gene expression data of five and 24 pairs, respectively, of endometrial tissues of pregnant women and patients with RIF, were downloaded from the Gene Expression Omnibus (GEO) database. Based on the similarity of lncRNA expression levels, a clustering heatmap was generated using the heatmap package in R software. The differentially expressed lncRNAs (DElncRNAs) between the RIF and control groups were identified based on the following criteria: |logFC|>1.5 and $p < 0.01$. The DElncRNAs of the two datasets were intersected to obtain the key lncRNAs that regulate the RIF process.

## Evaluation of FLI1-*PART1* binding

The interaction of FLI1 with *PART1* in endometrial epithelial cells was examined using the chromatin immunoprecipitation (ChIP) assay. Briefly, cross-linking and cracking

were performed to stabilize the complex. The nuclear components were separated with anti-FLI1 or anti-IgG antibodies. Chromatin fragmentation was performed, and protein-DNA complexes were captured and subjected to immunoprecipitation. Next, cross-linking and DNA purification were performed. The differential enrichment of *PART1* between different groups was analyzed. The dual-luciferase reporter assay was used to further verify the interaction between FLI1 and *PART1*.

## Cell lines and cell culture

The non-receptive endometrial epithelial cell line (HEC-1-A) and the receptive endometrial epithelial cell lines (Ishikawa and RL95-2) were purchased from Wuhan Punosai Life Technology Co., Ltd (Hubei, China). These cell lines have stable phenotypes and can be used to effectively simulate primary endometrial epithelial cells *in vitro*. BeWo cells were sourced from Wuhan Punosai Life Technology Co., Ltd (Hubei, China). Cell lines were cultured at 37 °C and 5% $CO_2$ in an incubator with saturated humidity. The cells were subcultured, and logarithmic phase cells were used for subsequent cell experiments.

## Construction of stable FLI1-overexpressing and *FLI1* knockdown cell lines

The FLI1 overexpression plasmid (OE-FLI1) and the corresponding negative quality control plasmid (technical control, OE-NC) were constructed by Shanghai Hanheng Biotechnology Co., Ltd (Shanghai, China). Meanwhile, the *FLI1* knockdown plasmid (sh-FLI1) and the corresponding negative quality control plasmid (technical control, sh-NC) were constructed by Guangzhou Huijun Biotechnology Co., Ltd (Guangdong, China).

The sequences of sh-FLI1 and sh-NC are as follows:

5′-CCGGCGTCATGTTCTGGTTTGAGATCTCGAGATCTCAAACCAGAACATGACG TTTTTGAATT-3′ (sh-FLI1#1);

5′-CCGGCCCATGAACTACAACAGCTATCTCGAGATAGCTGTTGTAGTTCATGGG TTTTTGAATT-3′ (sh-FLI1#2);

5′-CCGGCCCTTCTGACATCTCCTACATCTCGAGATGTAGGAGATGTCAGAAGGG TTTTTGAATT-3′ (sh-FLI1#3);

5′-CCGGTCCTAAGGTTAAGTCGCCCTCGCTCGAGCGAGGGCGACTTAACCTTAGG TTTTTGAATT-3′ (sh-NC).

Logarithmic phase Ishikawa and HEC-1-A cells were randomly divided into the following four groups: OE-NC group, transfected with OE-NC construct; OE-FLI1 group, transfected with OE-FLI1 construct; sh-NC group, transfected with sh-NC construct; sh-FLI1 group, transfected with sh-FLI1 construct. Transfection was performed using Lipo8000™ (Beyotime Biotechnology, Shanghai, China). At 24 h post-transfection, the cell line stably expressing the target vectors was obtained.

## Cell function assays

The effect of FLI1 on the proliferation of endometrial epithelial cells was examined using the cell counting kit-8 (CCK-8) kit (Beyotime Biotechnology, Shanghai, China), following the manufacturer's instructions. Meanwhile, the effect of FLI1 on the migration of endometrial
epithelial cells was evaluated using the scratch assay. Image J (National Institutes of Health, Bethesda, USA) was used to calculate the scratch area. The migration rate was calculated as follows: migration rate (%) = [(scratch area at 0 h –scratch area at 48 h)/(scratch area at 0 h) ×100%]. The effect of FLI1 on the epithelial-to-mesenchymal transition of endometrial epithelial cells was examined by evaluating the protein expression levels of vimentin and E-cadherin using western blotting. Furthermore, the effect of FLI1 on embryo adhesion was examined using the BeWo cells.

## Mouse model construction and adeno-associated virus (AAV) infection

Sexually mature male and female ICR mice (aged 8 weeks) of specific pathogen-free (SPF) grade were purchased from Skbex Biotechnology Co., Ltd. (Heinan, China). The males were fertile. The animals were caged for mating in a ratio 1:2 (male:female). Pregnancy in female mice was confirmed through vaginal suppositories. Pregnant mice were selected for subsequent *in vivo* experiments.

Epithelial cell-specific *PART1* overexpression AAV vector (AAV-PART1) and its corresponding negative control vector (AAV-NC) were constructed and packaged by Shanghai Hanheng Biotechnology Co., Ltd. (Shanghai, China). The virus titer was $1.00 \times 10^{12}$ VG/mL. AAV-NC and AAV-PART1 were injected into the mouse uterine cavity through the uterine horn. On days 0, 2, 5, and 8 of injection, endometrial epithelial cells were isolated to examine the overexpression efficiency of AAV-PART1 using qRT-PCR and western blotting.

## AAV injection

Twelve pregnant mice were randomly divided into the following two groups (6 mice/group): AAV-NC and AAV-PART1 groups. This study was approved by the Ethics Committee at Zhengzhou University (No. ZZU-LAC20200828[08]). All mice were housed in an SPF animal room under the following conditions: temperature, 24–26 °C; humidity, 50%–60%; circadian cycle, 12-h light/dark cycle (lights on from 6:00 AM to 6:00 PM); feed, irradiated feed; water, sterile water; environmental enrichment, each cage contained nesting material and chew stick. At day 1.5 of pregnancy, pregnant mice were anesthetized with isoflurane inhalation and placed prone on a small animal operating table insulated at 37 °C. The skin on the mouse back was fully exposed. A small incision was introduced on the back to fully expose the uterine horn. An insulin syringe was used to inject 20 $\mu$L of AAV into the uterine horn (mice in the AAV-NC and AAV-PART1 groups were injected with AAV-NC and AAV-PART1 constructs, respectively). The wound was sutured, and the animal was placed back in the animal room for separate feeding after it regained consciousness. On day 9 of pregnancy, blue dye was injected into the tail vein, and the mice were euthanized using a $CO_2$ euthanasia chamber. The chamber was initially filled with a mixture of $CO_2$ and $O_2$ in a ratio 6:4. The $CO_2$ concentration was gradually increased to 100% after the animals lost consciousness. The mice were maintained in the chamber for a minimum of 10 min to confirm death during which they were completely unconscious. None of the animals survived at the end of the study. The uterus was removed to observe the number of implanted embryos.

## Statistical analysis

All data are represented as mean ± standard deviation. GraphPad Prism 6.0 was used for statistical analysis and plotting. Means between two groups were compared using the Student's $t$-test or Fisher's exact test, while those between multiple groups were compared using one-way analysis of variance, followed by Dunnett's test. Differences were considered significant at $p < 0.05$.

# RESULTS

## Single-cell clustering analysis and identification of cells in the endometrial tissue

Single-cell sequencing data were subjected to cluster analysis and cell gene labeling to identify the following eight types of cell populations in the endometrial tissue: stromal fibroblast cells, endometrial cells, epithelial cells, T cells, B cells, dendritic cells (DCs), smooth muscle cells, and neural progenitor cells (Fig. 1A). The cell populations of the RIF and control group were subjected to t-SNE nonlinear dimensionality reduction analysis. Seven cell types were identified in both RIF and control groups with stromal fibroblast cells accounting for the highest proportion (Fig. 1B). Furthermore, heatmaps were used to display the top 10 differentially expressed genes of seven cell types (Fig. 1C). The key genes in each cell group were analyzed, and these genes can serve as biomarkers for identifying cell types. *CDH11*, *SPINK4*, *DMD*, *CD247*, *CD70*, *CD83*, *ITGA8*, and *UBE2C* were biomarkers for stromal fibroblasts, endometrial cells, epithelial cells, T cells, B cells, DCs, smooth muscle cells, and neural progenitor cells, respectively (Figs. 1D–1K).

## Endometrial epithelial cell numbers are downregulated and FLI1 expression is upregulated in patients with RIF

The frequency and proportion of different cell populations in the RIF and NC groups were statistically analyzed. Compared with that in the NC group, the proportion of interstitial fibroblasts and epithelial cells was significantly downregulated and the proportion of endothelial cells, T cells, and B cells was significantly upregulated in the RIF group. The difference in the proportion of epithelial cells between the RIF and NC groups was the highest (0.17% *vs* 20.94%, Fig. 2A).

ATAC-seq analysis of endometrial epithelial cells revealed that the chromatin accessibility was significantly different between the RIF and control groups. Most peaks were located at the enhancer position (enhancer was defined as the region that was >2 kb away from the transcription start site). The active enhancer histone-modified marker H3K27Ac was analyzed using ChIP-seq. Most of the upregulated peaks were enriched with H3K27Ac, suggesting that these peaks are mainly located in the active enhancer region, which may be related to gene transcription activation. These peaks were subjected to transcription factor (TF) motif analysis (MEME software). The most significantly enriched motif was FLI1, which was 65% enriched among all differentially upregulated peaks. Figure 2B shows the peak distribution of FLI1. Further analysis revealed that the chromatin accessibility of FLI1 in the endometrial epithelial cells of the RIF group was significantly higher than that in the

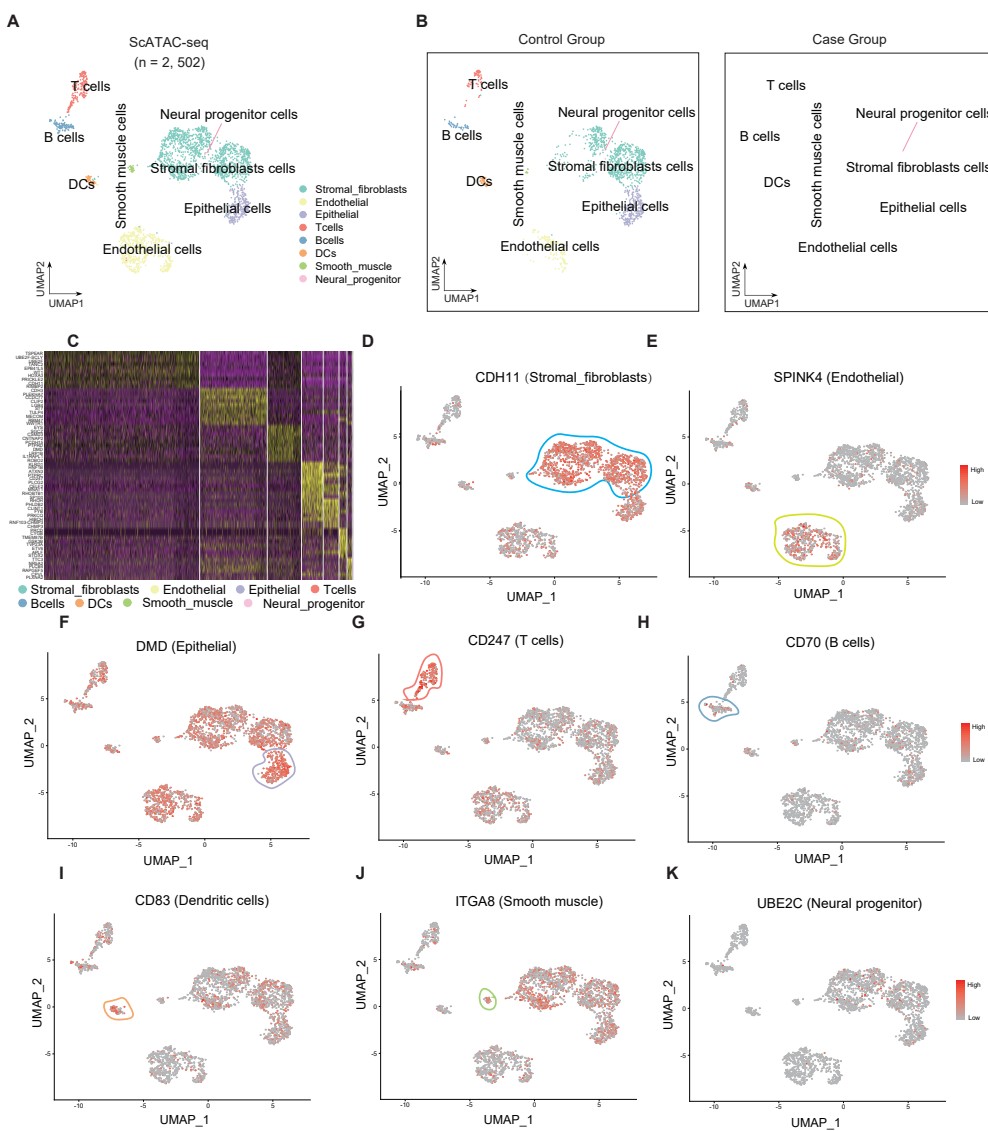

**Figure 1** **Single-cell clustering analysis and identification of cells in the endometrial tissue.** (A) Single-cell t-SNE clustering and identification of cells in the endometrial tissue. (B) Single-cell t-SNE cluster-ing and identification of cells in the endometrial tissue of the RIF (left) and control (right) groups; (C) Heatmap showing the relative expression levels of the top 10 genes in each cell group. Visualization of key genes in each cell group (D–K). t-SNE, t-distributed stochastic neighbor embedding; RIF, recurrent im-plantation failure; NC, control group.

endometrial epithelial cells of the control group (log FC = 0.27, $p = 0.009$). These findings suggest that FLI1 regulates embryo implantation.

The mRNA and protein expression levels of FLI1 in primary epithelial cells obtained from the endometrial tissues of the RIF and control groups were examined using qRT-PCR and western blotting. Compared with those in the control group, the endometrial cell mRNA and protein expression levels of FLI1 were markedly upregulated in the RIF group

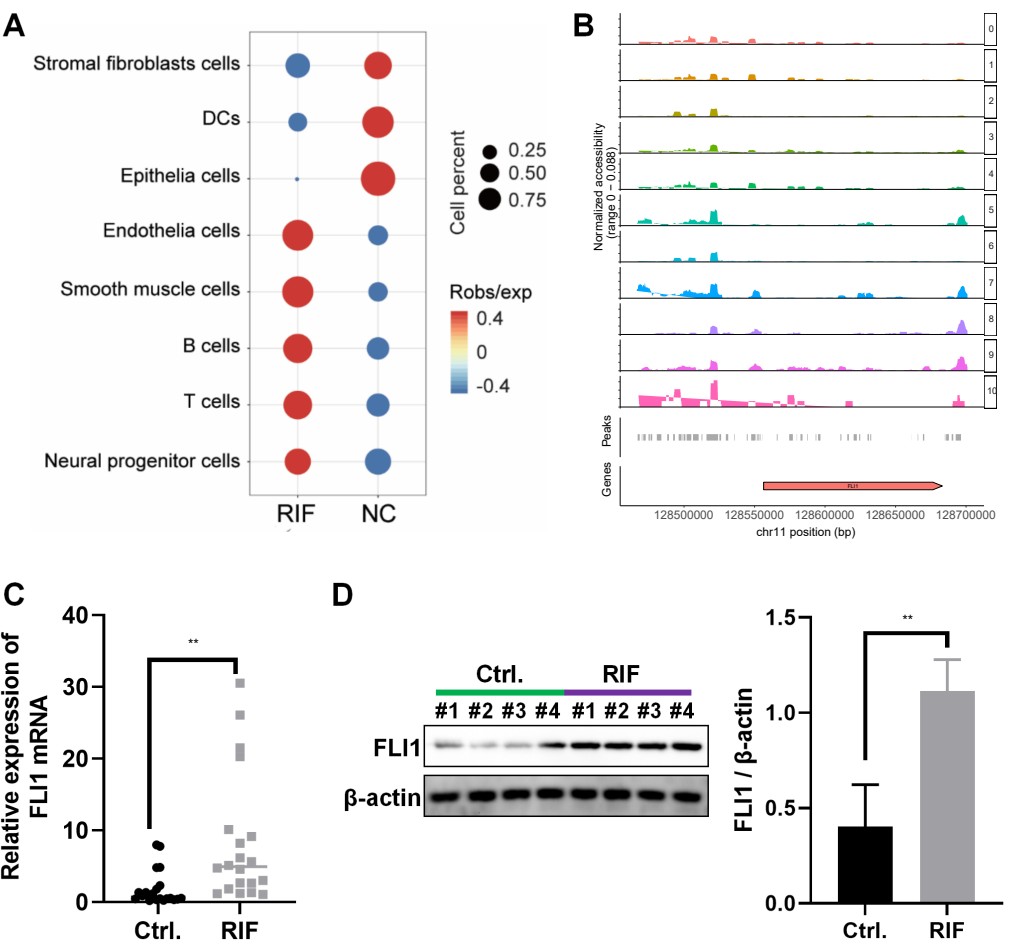

**Figure 2** **Decreased endometrial receptivity in patients with RIF is associated with downregulated endometrial epithelial cell content and aberrant FLI1 expression.** (A) The number and proportion of eight cell populations in the endometrial tissue of the RIF and control groups. (B) The scATAC-seq data were used to analyze the peaks of FLI1. The height of each color peak represents the opening degree of corresponding cell type chromatin; (C) The mRNA expression of *FLI1* in the endometrial epithelial cells was examined using qRT-PCR analysis. (D) The protein expression of FLI1 in endometrial epithelial cells was examined using western blotting. scATAC, single-cell assay for transposase accessible chromatin with high-throughput sequencing; NC, control group; qRT-PCR, quantitative real-time polymerase chain reaction; RIF, recurrent implantation failure; *p < 0.05; **p < 0.01; ***p < 0.001.

(Figs. 2C–2D). These data indicate that the chromatin accessibility of the TF FLI1 in endometrial epithelial cells of patients with RIF is enhanced. As the expression level of *FLI1* was significantly upregulated, it may be involved in inducing decreased endometrial receptivity in RIF.

## Effect of FLI1 on cell function *in vitro*

Non-receptive endometrial epithelial cell line (HEC-1-A cells) and receptive endometrial epithelial cell lines (both Ishikawa and RL95-2) were selected to perform the *in vitro* assays. qRT-PCR analysis revealed that the mRNA expression level of *FLI1* in HEC-1-A

cells (1.02 ±0.12) was significantly higher than that in Ishikawa (0.81 ±0.15) and RL95-2 cells (−3.91 ±0.50) ($p < 0.0001$, Fig. 3A). Thus, *FLI1* expression is upregulated in non-receptive endometrial epithelial cells. Stable transfection lines were established by transfecting FLI1 overexpression and knockdown plasmids into Ishikawa and HEC-1-A cells, respectively (Figs. 3B–3C). *FLI1* knockdown significantly increased the proliferation of HEC-1-A cells ($p < 0.0001$). Meanwhile, *FLI1* overexpression significantly inhibited the proliferation of Ishikawa cells ($p < 0.0001$, Fig. 3D). These findings indicate that the proliferation of endometrial epithelial cells is downregulated upon *FLI1* overexpression and upregulated upon *FLI1* knockdown. The results of the scratch test revealed that *FLI1* knockdown significantly increased the migration of HEC-1-A cells (Fig. 3E). Meanwhile, FLI1 overexpression significantly inhibited the migration of Ishikawa cells (Fig. 3F). Thus, the migration of endometrial cells is downregulated upon *FLI1* overexpression and upregulated upon *FLI1* knockdown. *FLI1* knockdown upregulated vimentin protein expression and downregulated E-cadherin protein expression in HEC-1-A cells (Figs. 3G–3H). Meanwhile, FLI1 overexpression significantly downregulated vimentin protein expression and upregulated E-cadherin protein expression in Ishikawa cells (Figs. 3G–3H). These findings indicate that the epithelial-to-mesenchymal transition of endometrial epithelial cells is downregulated upon *FLI1* overexpression and upregulated upon *FLI1* knockdown. The ability of FLI1-overexpressing Ishikawa cells to adhere to BeWo cells was significantly downregulated (Fig. 3I). In contrast, the ability of *FLI1* knockdown HEC-1-A cells to adhere to BeWo cells was significantly upregulated (Fig. 3I). These findings suggest that the ability of endometrial cells to adhere to BeWo cells was downregulated upon *FLI1* overexpression and upregulated upon *FLI1* knockdown.

## Effect of FLI1 on mouse embryo implantation

To determine the effect of *FLI1* on mouse embryo implantation, AAV-OE-NC and AAV-OE-FLI1 were injected into the uterus of female mice through the uterine horn. The FLI1 expression levels in the endometrial epithelial cells of each group of mice on days 0, 2, 5, and 8 post-injection were examined using qRT-PCR and western blotting. qRT-PCR analysis revealed that the *FLI1* mRNA expression levels were significantly upregulated in female mouse endometrial epithelial cells from day 2 post-AAV-OE-FLI1 injection (Fig. 4A). Consistent with the qRT-PCR analysis results, western blotting revealed that the expression level of FLI1 protein in female mouse endometrial epithelial cells was significantly upregulated from day 2 post-AAV-OE-FLI1 injection (Fig. 4B). AAV-OE-FLI1 injection into the uterine horn promoted FLI1 overexpression in mouse endometrial epithelial cells. Furthermore, the number of implanted embryos in the AAV-OE-FLI1-injected group was significantly lower than that in the AAV-OE-NC-injected group (Fig. 4C). These data indicate that epithelial cell-specific overexpression of *FLI1* can inhibit mouse embryo implantation.

## Molecular mechanism underlying the inhibitory effects of FLI1 on embryo implantation

In the GSE26787 dataset, 22 differentially expressed lncRNAs (DElncRNAs) were identified between the paired endometrial tissues of pregnant women and patients with RIF (13

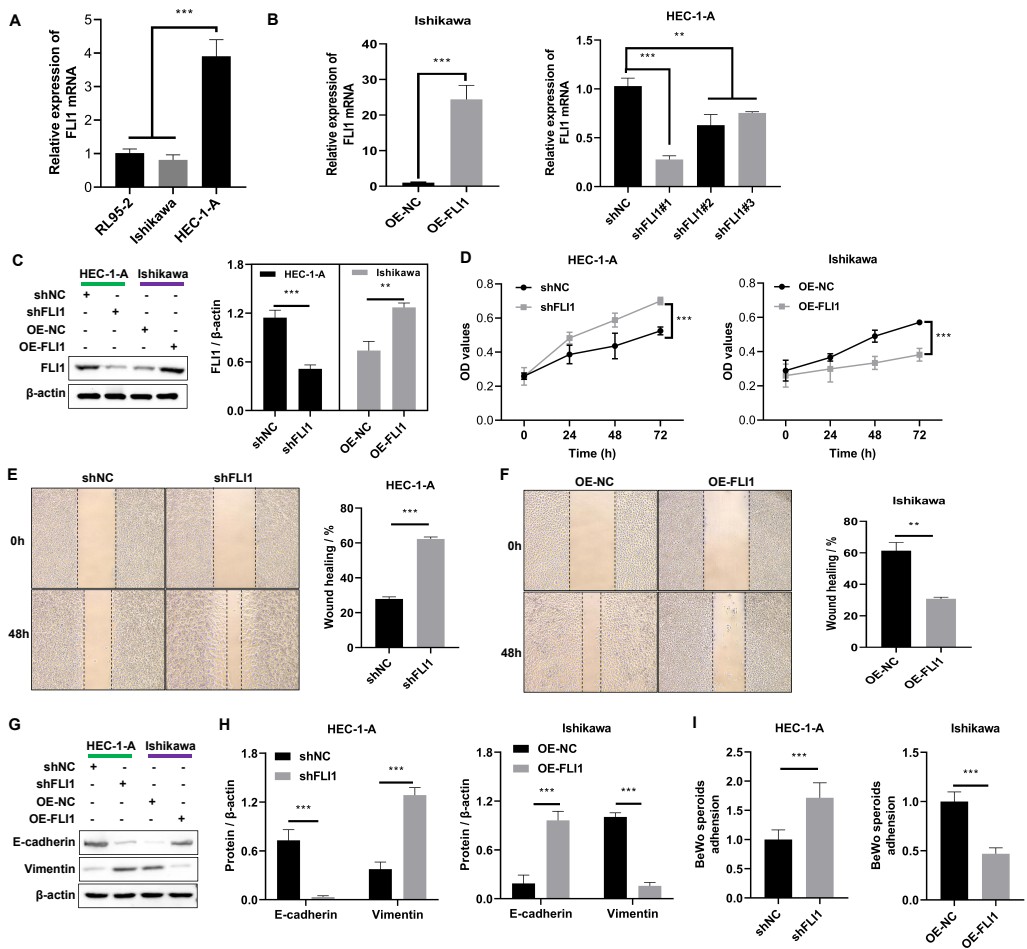

**Figure 3 Effect of FLI1 on cell functions.** (A) The *FLI1* expression levels in HEC-1-A, Ishikawa, and RL95-2 cells were examined using qRT-PCR analysis. Verification of the efficiency of FLI1 overexpression and silencing plasmid stable transfection using qRT-PCR (B) and western blotting (C) in HEC-1-A and Ishikawa cells. (D) The proliferation of HEC-1-A and Ishikawa cells transfected with OE-NC and OE-FLI1 plasmids was examined using the CCK-8 assay. The invasive ability of HEC-1-A (E) and Ishikawa (F) cells transfected with OE-NC and OE-FLI1 plasmids was analyzed using the scratch test. The protein expression of vimentin and E-cadherin in HEC-1-A and Ishikawa cells transfected with OE-NC and OE-FLI1 plasmids was examined using western blotting (G–H). (I) The effect of FLI1 overexpression on the ability of Ishikawa cells to adhere to BeWo cells and the effect of *FLI1* knockdown on the ability of HEC-1-A cells to adhere to BeWo cells. OE-FLI1 and OE-NC, FLI1 overexpression plasmid and its negative control plasmid, respectively; sh-FLI1 and sh-NC, *FLI1* knockdown plasmid and its negative control plasmid; NC, control group; qRT-PCR, quantitative real-time polymerase chain reaction; CCK-8, cell counting kit-8; OE, overexpression. $*p < 0.05$; $**p < 0.01$; $***p < 0.001$; $****p < 0.0001$.

upregulated lncRNAs and nine downregulated lncRNAs) (Fig. 5A). Meanwhile, in the GSE111974 dataset, 72 DElncRNAs were identified between the paired endometrial tissues of pregnant women and patients with RIF (55 upregulated lncRNAs and 17 downregulated lncRNAs) (Fig. 5B). *LOC100505912* and *PART1* were the only common DElncRNAs between GSE26787 and GSE111974 datasets. The expression levels of *LOC100505912* and *PART1* were significantly upregulated in the endometrial tissue of patients with RIF

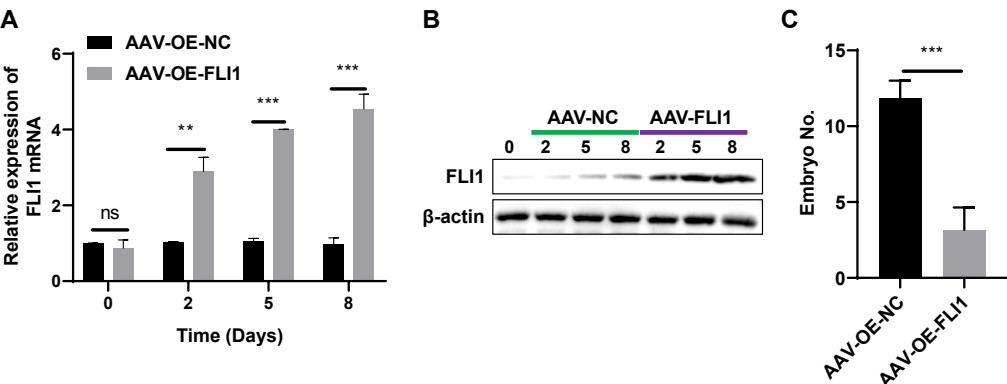

**Figure 4 Epithelial cell-specific FLI1 overexpression inhibits mouse embryo implantation.** The expression levels of FLI1 in the mouse endometrial epithelial cells were analyzed using qRT-PCR and western blotting (A and B). (C) The effect of epithelial cell-specific FLI1 overexpression on the number of implanted embryos in pregnant mice. AVV-OE-FLI1 and AVV-OE-NC, adeno-associated virus vector encoding FLI1 overexpression plasmid and its negative control plasmid, respectively. qRT-PCR, quantitative real-time polymerase chain reaction. **$p < 0.01$; ***$p < 0.001$.

(Fig. 5C). Therefore, we hypothesized that the dysregulated *LOC100505912* and *PART1* in the endometrial tissue are the key lncRNAs regulating RIF.

qRT-PCR analysis revealed that the expression of *PART1* was significantly upregulated in FLI1-overexpressing endometrial epithelial cells and significantly downregulated in *FLI1* knockdown endometrial epithelial cells (Fig. 6A), indicating that *FLI1* regulates *PART1* expression. However, *FLI1* overexpression or knockdown in endometrial epithelial cells did not affect the expression level of *LOC100505912* (Fig. 6B). Meanwhile, analysis with the JASPAR database (https://jaspar.genereg.net/) revealed that FLI1 had binding sites in the *PART1* promoter region with a score of 0.86 (Fig. 6C) but not in the *LOC100505912* promoter.

The binding of FLI1 to *PART1* was examined using the ChIP and dual-luciferase reporter assays. The ChIP assay results revealed that the enrichment level of *PART1* in the anti-FLI1 group was significantly higher than that in the anti-IgG group (Fig. 6D), indicating the interaction between FLI1 and *PART1* promoter. The results of the dual-luciferase reporter assay demonstrated that the wild-type *PART1* promoter, but not the mutant *PART1* promoter, enhanced the luciferase reporter activity of FLI1 (Fig. 6E). These data indicated that FLI1 directly promotes *PART1* transcription by binding to its promoter.

To determine the effect of *PART1* on mouse embryo implantation, AAV-OE-NC and AAV-OE-PART1 were injected into the uterus of pregnant mice through the uterine horn. The *PART1* expression levels in the endometrial epithelial cells of different groups were examined using qRT-PCR analysis on day 8 post-injection. Injection with AAV-OE-PART1 significantly upregulated the expression level of *PART1* mRNA in female mouse endometrial epithelial cells (Fig. 7A). Furthermore, the number of implanted embryos in

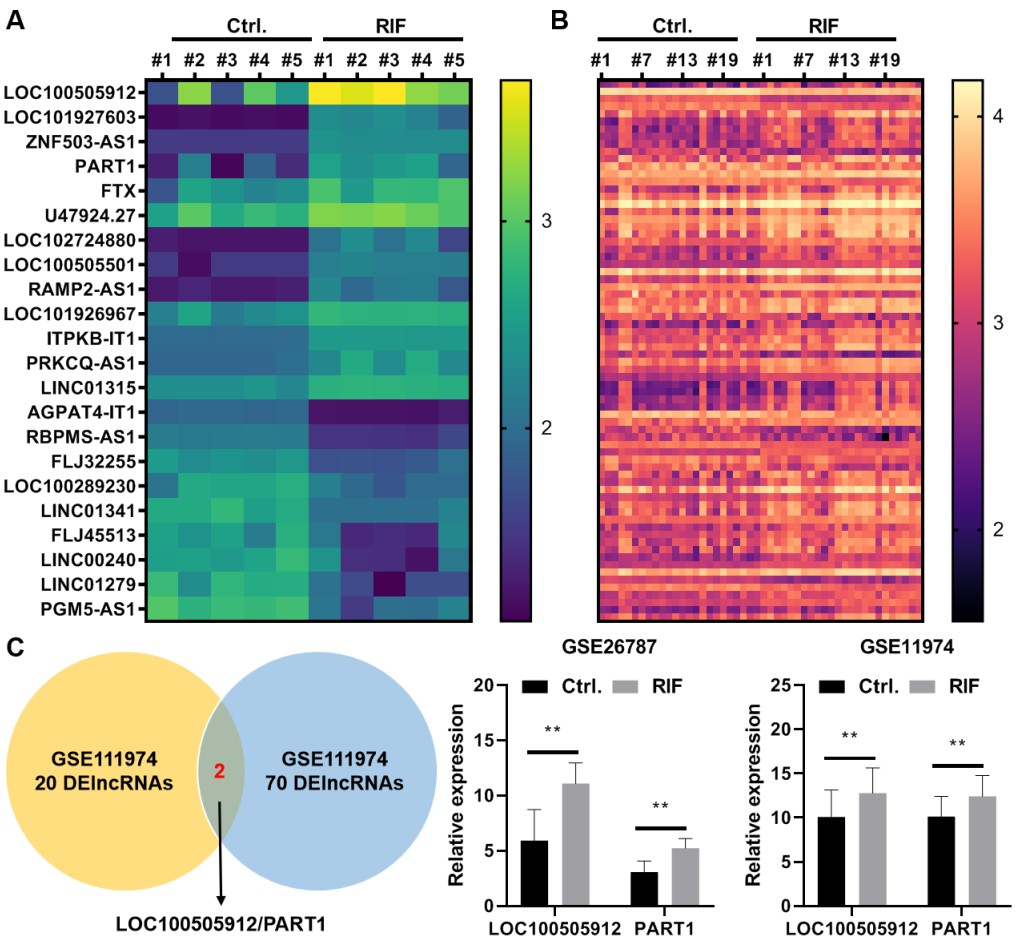

**Figure 5** *LOC100505912* and *PART1* are upregulated in the endometrial tissue of patients. (A) Heatmap showing the differentially expressed long non-coding RNAs (lncRNAs) in the GSE26787 dataset. (B) Heatmap showing the differentially expressed lncRNAs in the GSE111974 dataset. (C) The Venn plot and bar plot showing the common differentially expressed lncRNAs between the GSE26787 and GSE111974 datasets. Ctrl and RIF, endometrial tissues of healthy pregnant women and patients with recurrent implantation failure, respectively; DElncRNAs: differentially expressed lncRNAs. ** $p < 0.01$.

the AAV-OE-PART1-treated group was significantly lower than that in the AAV-OE-NC-treated group (Fig. 7A). Thus, epithelial cell-specific overexpression of PART1 can inhibit mouse embryo implantation.

The results of the cell function experiments revealed that the proliferation of Ishikawa cells was significantly downregulated upon FLI1 overexpression and significantly upregulated upon *PART1* knockdown (Fig. 7B). The scratch test results revealed that the migration of Ishikawa cells was significantly downregulated upon *FLI1* overexpression and significantly upregulated upon *FLI1* knockdown (Fig. 7C). Western blotting demonstrated that FLI1 overexpression downregulated the vimentin levels and upregulated the E-cadherin levels in Ishikawa cells, whereas *PART1* knockdown exerted the opposite effects (Fig. 7D). FLI1 overexpression inhibited the epithelial-to-mesenchymal transition of endometrial

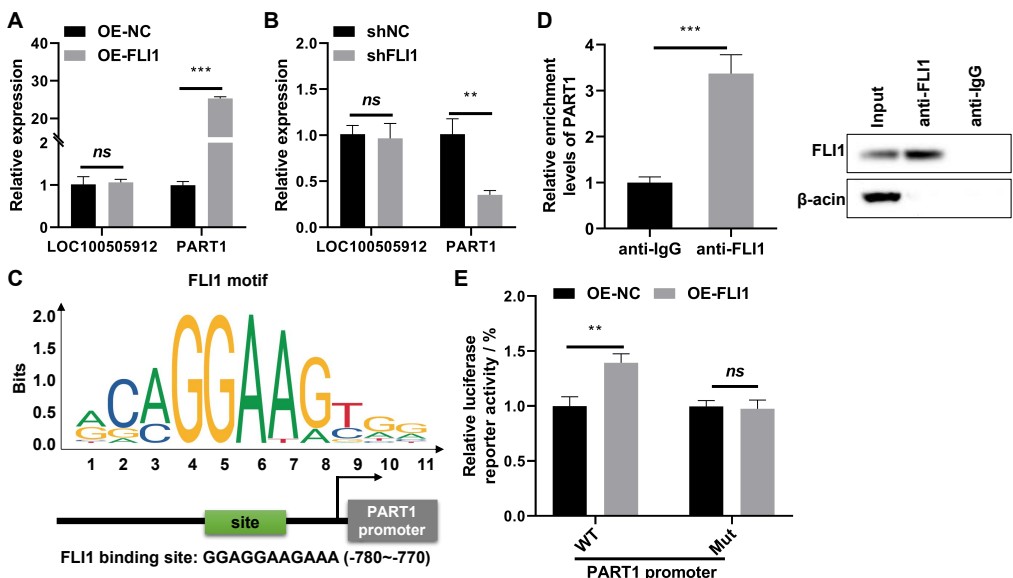

**Figure 6** **FLI1 promotes the transcription of *PART1* by binding to its promoter.** qRT-PCR analysis of the effect of FLI1 overexpression or knockdown on the *PART1* and *LOC100505912* levels in endometrial epithelial cells (A–B). (C) The binding site of FLI1 in the *PART1* promoter was predicted using JASPAR. (D) The interaction between FLI1 and *PART1* promoter was examined using the ChIP assay. (E) The results of the dual-luciferase reporter assay demonstrated that FLI1 regulates *PART1* transcription by binding to its promoter. sh-FLI1 and sh-NC, FLI1 knockdown plasmid and its negative control plasmid, respectively; OE-FLI1 and OE-NC, FLI1 overexpression plasmid and its negative control plasmid, respectively; qRT-PCR, quantitative real-time polymerase chain reaction; ChIP, chromatin immunoprecipitation. $**p < 0.01$; $***p < 0.001$.

epithelial cells by upregulating *PART1* expression. The ability of Ishikawa cells to adhere to BeWo cells was significantly downregulated upon FLI1 overexpression and significantly upregulated upon *PART1* knockdown (Fig. 7E).

These results indicate that the binding of FLI1 to the *PART1* promoter promotes the transcription and expression of *PART1*. Epithelial cell-specific *PART1* overexpression inhibited mouse embryo implantation and endometrial cell proliferation, invasion, epithelial-to-mesenchymal transition, and adherence to BeWo cells. Thus, *PART1* contributes to decreased endometrial receptivity and embryo implantation failure.

## DISCUSSION

RIF is associated with diverse etiological factors, such as embryonic genome defects, endometrial aberrations, maternal autoimmune factors, and sperm quality. Endometrial aberrations are one of the common causes of RIF. The endometrium is an important site for embryo implantation and development. Hence, the structural and functional aberrations in the endometrium directly affect the success rate of implantation. Previous studies have revealed the role of endometrial aberrations in RIF. Endometriosis can lead to dysregulated proliferation and differentiation of endometrial cells, resulting in

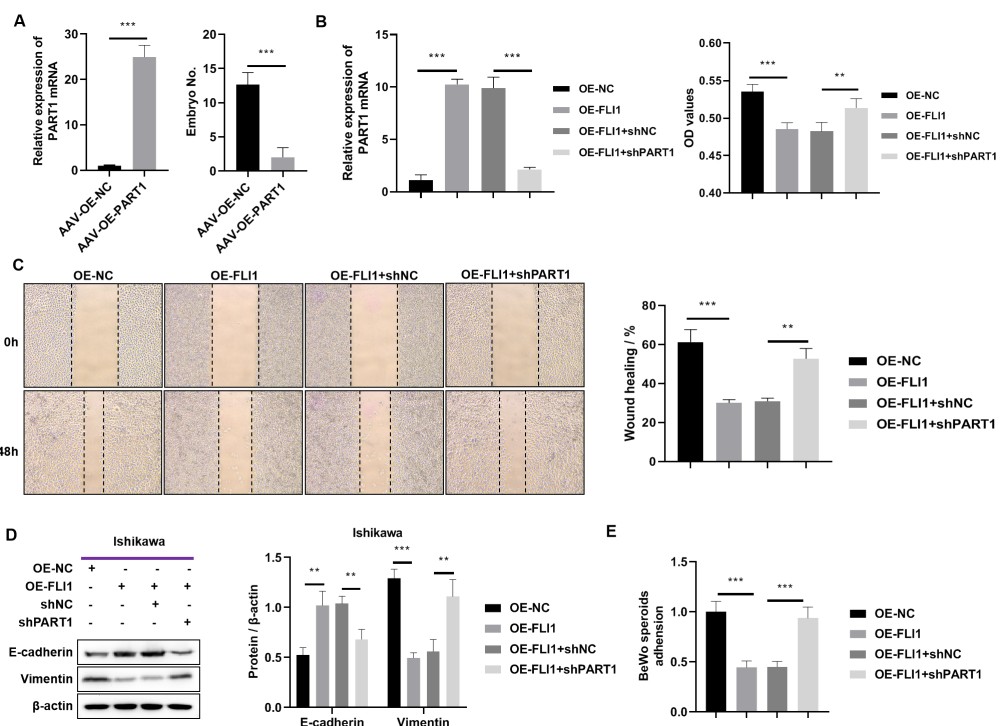

**Figure 7** **Effects of epithelial cell-specific PART1 overexpression on mouse embryo implantation and endometrial epithelial cell function.** (A) The mRNA expression level of *PART1* in the mouse endometrial epithelial cells and the effect of PART1 overexpression on the number of implanted embryos in pregnant mice. (B) The expression level of PART1 in the endometrial epithelial cells and the analysis of endometrial epithelial cell proliferation in different groups using the CCK-8 assay. (C) Cell invasion ability was examined using the scratch test. (D) Western blotting analysis of vimentin and E-cadherin in Ishikawa cells; (E) Cell adhesion ability was examined using the BeWo cells. AVV-OE PART1 and AVV-OE-NC, adeno-associated virus encoding PART1 overexpression construct and its negative control construct, respectively; OE-FLI1 and OE-NC, FLI1 overexpression plasmid and its negative control plasmid, respectively; sh-PART1 and sh-NC, *PART1* knockdown plasmid and its negative control plasmid, respectively; CCK-8, cell counting kit-8; **$p < 0.01$; ***$p < 0.001$.

the formation of ectopic foci and polyps, hindering embryo implantation (*Miravet-Valenciano et al., 2017*). Additionally, endometriosis can promote uterine wall stiffness and narrowing of the lumen, which adversely affect implantation. The increased thickness of the endometrium, which can lead to RIF, can impair communication between the embryo and the mother, affecting embryo implantation. The decreased thickness of the endometrium is not conducive to embryo implantation and growth (*Vartanyan, Tsaturova & Devyatova, 2020*). Endometrial infection and inflammation can lead to aberrant local immune responses in the endometrium and consequently prevent embryo implantation (*Wang et al., 2022*). Furthermore, the dysregulation of endometrial microbiota has been associated with implantation failure (*Lozano et al., 2023*). Endometrial receptivity refers to the ability of the endometrium to allow the localization, adhesion, penetration, and implantation of blastocysts, as well as to enable embryo implantation and development (*Lessey & Young, 2019*). Additionally, endometrial receptivity is closely related to embryo

implantation and is one of the key factors determining the success of ART (*Bui, Timmons & Young, 2022*). Currently, limited methods are available for treating endometrial receptivity aberrations. This is because the molecular mechanism underlying impaired endometrial receptivity-induced RIF has not been elucidated. This study examined the molecular mechanism underlying impaired endometrial receptivity-induced RIF using scATAC-seq technology and *in vitro* and *in vivo* studies.

scATAC-seq and clinical sample analyses revealed the decreased proportion of endometrial epithelial cells and the upregulation of *FLI1* expression in the endometrial epithelial tissue of patients with RIF (Fig. 2). Single-cell sequencing can reveal the gene structure and expression profile of individual cells, providing valuable insights into structural and copy number variations and RNA expression levels. Additionally, single-cell sequencing enables the precise differentiation of cell types and facilitates the elucidation of molecular mechanisms at the cellular level. A previous study performed scRNA-seq and reported that the expression of endometrial receptivity-related genes in four major endometrial fibroblast-like cells of patients with RIF was distinct from that in four major endometrial fibroblast-like cells of control subjects (*Ruane et al., 2022*). The proportion of CD49a+CXCR4+ NK cells was downregulated in patients with RIF. In this study, scATAC-seq analysis revealed that the number of endometrial epithelial cells was significantly downregulated in the endometrial tissue of patients with RIF (Fig. 2A). Analysis of the peaks of FLI1, which were examined using scATAC-seq data, revealed that FLI1 exerted significant regulatory effects on embryo implantation (Fig. 2B). Endometrial epithelial cells, which are the main components of the endometrium, play a critical role in embryo implantation and mid-term pregnancy (*Lai et al., 2022*; *Singh & Aplin, 2009*). After ovulation, endometrial epithelial cells secrete increased amounts of mucus to provide a favorable environment for embryo transfer. Furthermore, endometrial epithelial cells secrete essential nutrients, such as glucose and amino acids, which promote embryonic development. Additionally, endometrial epithelial cells regulate the immune response in the endometrium by secreting various bioactive molecules, such as cytokines and chemokines, preventing the immune system from rejecting the embryo. Moreover, endometrial epithelial cells facilitate embryo implantation and ensure successful attachment by secreting growth factors, adhesion molecules, and other compounds. Progesterone promotes the synthesis and secretion of hormones (like progesterone), which support pregnancy and fetal growth and development, in the endometrial epithelial cells. The ETS family of TFs regulates cell proliferation, apoptosis, differentiation, and migration (*He et al., 2021*). Among the members of this family, the role of FLI1 in various physiological and pathological processes has piqued the interest of the scientific community. FLI1 regulates physiological hematopoiesis, vasculogenesis, immune response, and pro-fibrotic processes (*Mikhailova et al., 2023*). However, the role of FLI1 in endometrial receptivity and RIF has not been previously examined. In this study, *FLI1* was upregulated in non-receptive endometrial epithelial cells. Additionally, FLI1 overexpression inhibited endometrial epithelial cell proliferation, migration, stromal transformation, and binding to BeWo cells (Fig. 3), which are critical processes for embryo transfer and endometrial receptivity. For example, the epithelial-to-mesenchymal transition can mediate the growth of trophoblast, which
is essential for the physiological function of the endometrium and the implantation and development of an embryo (*Crha et al., 2019*). Cell adhesion molecules are reported to be critical for human embryo implantation (*Fukuda & Sugihara, 2012*). Furthermore, *in vivo* experiments demonstrated that epithelial cell-specific FLI1 overexpression can inhibit mouse embryo implantation (Fig. 4). Therefore, these findings indicate that epithelial cells play an important role in endometrial receptivity in RIF. FLI1 is specifically expressed in the endometrial epithelial cells of patients with RIF and mediates endometrial receptivity by inhibiting serious of cell ability.

LncRNAs are non-coding RNA involved in the occurrence and development of several pathological conditions, including RIF (*Chen et al., 2019*; *Huang et al., 2021*; *Maduro, 2019*). Genome-wide analysis of lncRNA expression patterns in women with RIF revealed 148 lncRNAs corresponding to 147 cis-regulatory target genes. The cis-regulated target genes of these significant DElncRNAs are enriched in various pathways, including the tumor necrosis factor, Toll-like receptor, and NF- $\kappa$B signaling pathways (*Chen et al., 2019*; *Xu et al., 2019*). Next, the molecular mechanisms through which FLI1 affects RIF development and endometrial receptivity were examined. Preliminary analysis of GEO datasets demonstrated that the lncRNAs *PART1* and *LOC100505912* are involved in RIF development and endometrial receptivity (Fig. 5). Previous studies have reported that TFs modulate lncRNA transcription by binding to the lncRNA promoter region and consequently regulate the biological functions of cells (*Knauss et al., 2018*; *Zhang et al., 2022*). Therefore, FLI1 was hypothesized to regulate *PART1* and *LOC100505912* transcription by binding to their promoter regions. *FLI1* overexpression or knockdown in endometrial epithelial cells did not affect the expression level of *LOC100505912*. The JASPAR online tool was used to predict FLI1 binding sites in the *PART1* promoter region. The in silico findings were validated using the ChIP and dual-luciferase reporter assays, which revealed that FLI1 directly binds to the *PART1* promoter and facilitates its transcription (Fig. 6). *PART1* is a crucial factor involved in carcinogenesis and osteogenic differentiation of bone marrow mesenchymal stem cells as it regulates cell proliferation, apoptosis, invasion, and metastasis through various mechanisms (*Ran et al., 2022*; *Zhang et al., 2021*). However, the roles of *PART1* in RIF and endometrial receptivity have not been previously reported.

## CONCLUSIONS

This study revealed that *PART1* inhibited mouse embryo implantation and various biological processes of endometrial epithelial cells, such as proliferation, invasion, epithelial-to-mesenchymal transition, and adhesion to BeWo cells. Thus, the molecular mechanism of *FLI1* involves the upregulation of *PART1* expression, the inhibition of embryo implantation, and the suppression of endometrial receptivity in patients with RIF. These findings indicate that *FLI1* and *PART1* are potential therapeutic targets for RIF. However, this study focused only on the role of FLI1 and *PART1* in endometrial receptivity and RIF and did not examine other potential influencing factors. Additionally, this study was conducted using mice and cell lines. The clinical relevance of these findings must be validated using human samples.

## Funding
This work was supported by grants from the National Natural Science Foundation of China (32271169). The funders had no role in study design, data collection and analysis, decision to publish, or preparation of the manuscript.

## Grant Disclosures
The following grant information was disclosed by the authors:
The National Natural Science Foundation of China: 32271169.

## Competing Interests
The authors declare there are no competing interests.

## Author Contributions
- Yile Zhang conceived and designed the experiments, analyzed the data, prepared figures and/or tables, and approved the final draft.
- Beining Yin conceived and designed the experiments, analyzed the data, prepared figures and/or tables, authored or reviewed drafts of the article, and approved the final draft.
- Sichen Li performed the experiments, authored or reviewed drafts of the article, and approved the final draft.
- Yueyue Cui performed the experiments, authored or reviewed drafts of the article, and approved the final draft.
- Jianrong Liu conceived and designed the experiments, performed the experiments, authored or reviewed drafts of the article, and approved the final draft.

## Human Ethics
The following information was supplied relating to ethical approvals (i.e., approving body and any reference numbers):
The Ethics Committee at the First Affiliated Hospital of Zhengzhou University.

## Animal Ethics
The following information was supplied relating to ethical approvals (i.e., approving body and any reference numbers):
The Ethics Committee at Zhengzhou University.

## DNA Deposition
The following information was supplied regarding the deposition of DNA sequences:
The sequences are available at SRA: SRP435142; and BioProject: PRJNA963093.
Available at https://www.ncbi.nlm.nih.gov/bioproject/PRJNA963093.
Available at https://trace.ncbi.nlm.nih.gov/Traces/?view=study&acc=SRP435142.

## Data Availability
The data is available at Figshare: Zhang, Yile (2023). Raw data.zip. figshare. Journal contribution. Available at https://doi.org/10.6084/m9.figshare.22688389.v3.

## Supplemental Information

Supplemental information for this article can be found online at http://dx.doi.org/10.7717/peerj.16105#supplemental-information.

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
