# Peer review of "Friend leukemia integration 1 overexpression decreases endometrial receptivity and induces embryo implantation failure by promoting PART1 transcription in the endometrial epithelial cells"

_PeerJ, doi:10.7717/peerj.16105_

## Round 0.1 · original submission · Major Revisions

In view of the constructive comments of the reviewers, I suggest that you revise the manuscript according to these comments and give a reply one by one.

Reviewer 1 ·

Basic reporting

The study found that the FLI1 gene is specifically expressed in the endometrial epithelial cells of patients with recurrent implantation failure (RIF). Overexpression of FLI1 inhibits embryo implantation ability, while silencing of FLI1 has the opposite effect. This is an interesting study that requires some detailed revisions before publication.

First, some grammar, spelling and sentence structure should be proofread by fluent English speakers. Strengthening English expression can help readers understand the article more clearly.

Second, In Introduction, considering that the main viewpoint of this study is to explore the role and mechanism of FLI1 in endometrial receptivity and RIF, the author should supplement relevant literature and explain why FLI1 was chosen for the study in the manuscript. It is worth affirming that the author's choice of other references is reasonable, and it is obvious that the proportion of literatures in the past five years is very high.

Thirdly, in terms of article structure, I strongly recommend revising the conclusion as a separate chapter, rather than integrating it into the discussion now.

Fourthly, the figures and tables are generally standardized, and the original data can also be trusted. However, the font size in the figure should be as consistent as possible.

Experimental design

-

Validity of the findings

-

·

Basic reporting

Recurrent implantation failure (RIF) is the biggest obstacle to the success of IVF-ET, and has become a major problem to be solved urgently in assisted reproduction.This study found that FLI1 overexpression in RIF endometrial epithelial cells inhibited embryo implantation ability by binding to the PART1 promoter region to promote PART1 expression. These findings may lead to the development of new targets for treating RIF. The manuscript has a certain degree of innovation and can basically help readers understand the intention. Overall, it has a certain value, but the English writing of the manuscript needs to be improved.

In addition, the selection of references is reasonable, but their format does not comply with PeerJ's specifications. Please check.

Experimental design

1. In Abstract, the author indicated that “The purpose of this study is to explore the role and mechanism of FLI1 in endometrial receptivity and RIF.” The author should illustrate why FLI1 was selected for study in the manuscript.

2.A detailed description is required for the methods in Abstract instead of Cell experiments and in vivo ICR mice experiments were conducted to investigate the effects of FLI1 on embryo implantation ability. In addition, the results of the recovery experiment need to be summarized in the Abstract.
3. In Introduction, the authors should indicate that why is lncRNA being studied as a downstream regulatory factor. Of FLI1. In addition, the application of scRNA seq in IVF-ET research needs to be described.
4. In Materials and methods, FLI1 expression levels were assessed using real-time quantitative PCR (RT-qPCR) and Western blot (WB) analysis. Except for FLI1, the expression of Vimentin protein and E-cadherin protein was also detected. Therefore, the detailed method steps of RT-qPCR and WB need to be described.

Validity of the findings

1. Some results require a detailed description, such as TSNE nonlinear dimensionality reduction analysis was performed on the cell populations in the RIF and control groups, and the visualized results are exhibited in Figure 1B. Furthermore, heat maps were used to display the top 10 differentially expressed genes in each cell group (Figure 1C). The double Luciferase reporting experiment further proved that FLI1 binded to the PART1 promoter to promote PART1 transcription (Figure 6E).
2. The discussion needs to summarize the findings of the research combined with previous research, and identify the highlights of the research. In this study, the author provided extensive descriptions of the results in the discussion section. Therefore, the discussion requires a systematic and logical discussion based on the research purpose and findings.

Additional comments

Although the author has done a lot of work, one limitation of the study is that it focused primarily on in vitro and in vivo animal models, and the results may not fully reflect the complexity of the human reproductive system.

·

Basic reporting

The manuscript needs to be thoroughly revised for the adequate use of the English language. Word choices need to be revised.
There is not sufficient background literature referenced in the introduction and in the discussion section. Only broad concepts concerning RIF are provided, many of them not directly related to the focus of the manuscript. Insufficiently deep discussion is observed. There little to no information on previous research on the main molecules studied, FLI1 and PART1, and on how they may be impairing endometrial function.
ChIP assay data does not include blots and appropriate controls. If different experimental approach was used, please clarify.
Figures 1C and 2B do not present adequate resolution. It is difficult to read and interpret the images.
Abbreviations and acronyms used in figures should be described on legends. Some are missing on the current legend’s format.

Authors should revise the interval of pvalues indicated by ## and ###.

There is lack of connection between the abstract, introduction and supporting results. The manuscript includes a high-throughput approach, from which FLI1 is identified and then its relevance and mechanism investigated.
Labelling of the ladders on the raw blots are necessary. There seems to be different ladders used on different blots of the same protein.

Experimental design

The research question, which is better described in the introduction section, is relevant and meaningful. Results almost perfectly support a negative relationship between FLI1/PART1 expression and endometrial function, and the mechanistic relationship between FLI1 and PART1.

In overexpression and inhibition of expression experiments, a group of HEC and Ishikawa cells with no manipulation is missing.

Authors do not address the potential bias involved with assessing gene expression to compare one cell population that was significantly more abundant in normal patients than in RIF patients.

Dilution of antibodies used should be provided.

Beware that t test is not adequate for comparison among more than 2 groups. Specific statistical approach should be indicated for each group comparison.

Validity of the findings

Conclusions are supported by results. However, additional information are necessary regarding the raw data and controls for some experiments need to be shown.
Statistical approach should be better explained and indicated to which experiment it belongs.

---

## Round 0.2 · accepted · Accept

Both reviewers expressed satisfaction with your revisions and also made acceptance recommendations for your manuscript. Based on their full and rigorous opinions, I also made a decision to accept. Thank you for your recognition and support of PeerJ, and hope that you can submit more research to our journal in the future.

Reviewer 1 ·

Basic reporting

no comment

Experimental design

no comment

Validity of the findings

no comment

Additional comments

The authors' findings may lead to the development of new targets for treating RIF. I think the revision is sufficient. The revised version is acceptable.

·

Basic reporting

English has been improved and additional editing was performed, improving the manuscript.

Experimental design

Additional information regarding the methodology was provided.

Validity of the findings

Authors indicated the study's limitations, which does not reduces its relevance.

Additional comments

The authors addressed initial suggestions appropriately, therefore the manuscript is considered to be adequate for publication.